# Poor Decision Making and Sociability Impairment Following Central Serotonin Reduction in Inducible TPH2-Knockdown Rats

**DOI:** 10.3390/ijms25095003

**Published:** 2024-05-03

**Authors:** Lucille Alonso, Polina Peeva, Tania Fernández-del Valle Alquicira, Narda Erdelyi, Ángel Gil Nolskog, Michael Bader, York Winter, Natalia Alenina, Marion Rivalan

**Affiliations:** 1Institut für Biologie, Humboldt-Universität zu Berlin, 10099 Berlin, Germany; lucille.alonso@u-bordeaux.fr (L.A.); tania.fernandez@charite.de (T.F.-d.V.A.); york.winter@hu-berlin.de (Y.W.); 2Charité—Universitätsmedizin Berlin, Corporate Member of Freie Universität Berlin and Humboldt-Universität zu Berlin, 10117 Berlin, Germanymbader@mdc-berlin.de (M.B.); 3Univ. Bordeaux, CNRS, IINS, UMR 5297, F-33000 Bordeaux, France; 4Max Delbrück Center for Molecular Medicine in the Helmholtz Association (MDC), 13125 Berlin, Germany; 5DZHK (German Center for Cardiovascular Research), Partner Site Berlin, 10785 Berlin, Germany; 6Institute for Biology, University of Lübeck, 23562 Lübeck, Germany; 7NeuroPSI—Paris-Saclay Institute of Neuroscience, CNRS—Université Paris-Saclay, F-91400 Saclay, France

**Keywords:** tryptophan hydroxylase 2, TPH2, serotonin, tetracycline responsive system, inducible knockdown, vulnerability, network analysis, rat

## Abstract

Serotonin is an essential neuromodulator for mental health and animals’ socio-cognitive abilities. However, we previously found that a constitutive depletion of central serotonin did not impair rat cognitive abilities in stand-alone tests. Here, we investigated how a mild and acute decrease in brain serotonin would affect rats’ cognitive abilities. Using a novel rat model of inducible serotonin depletion via the genetic knockdown of tryptophan hydroxylase 2 (TPH2), we achieved a 20% decrease in serotonin levels in the hypothalamus after three weeks of non-invasive oral doxycycline administration. Decision making, cognitive flexibility, and social recognition memory were tested in low-serotonin (Tph2-kd) and control rats. Our results showed that the Tph2-kd rats were more prone to choose disadvantageously in the long term (poor decision making) in the Rat Gambling Task and that only the low-serotonin poor decision makers were more sensitive to probabilistic discounting and had poorer social recognition memory than other low-serotonin and control individuals. Flexibility was unaffected by the acute brain serotonin reduction. Poor social recognition memory was the most central characteristic of the behavioral network of low-serotonin poor decision makers, suggesting a key role of social recognition in the expression of their profile. The acute decrease in brain serotonin appeared to specifically amplify the cognitive impairments of the subgroup of individuals also identified as poor decision makers in the population. This study highlights the great opportunity the Tph2-kd rat model offers to study inter-individual susceptibilities to develop cognitive impairment following mild variations of brain serotonin in otherwise healthy individuals. These transgenic and differential approaches together could be critical for the identification of translational markers and vulnerabilities in the development of mental disorders.

## 1. Introduction

Mental health is a dynamic process and alternation between phases of deterioration and improvement of health is one hallmark of mental illness. In regard to the current lack of specific and universal treatments of mental disorders [1], identifying individual-specific targets to prevent a transition from adaptive to pathological mental states is essential [2]. Along the continuum between adaptive and maladaptive behavioral dimensions, vulnerable individuals at a higher risk for the development of psychiatric disorders would present a combination of interconnected preserved and impaired behaviors [3]. Following the network approach to mental disorder, a higher connectivity within such networks of symptoms is an objective characteristic of vulnerability to pathology [4]. In humans, making repeated poor decisions in everyday life is known to lead to long-term disadvantageous outcomes and a general deterioration of mental health. A poor decision-making ability is one common symptom of most human mental disorders [5]. In the Iowa Gambling Task, a test of decision making under everyday uncertain conditions of choices, approximately 30% of non-clinical populations present similar decision-making deficits as clinical populations [6,7,8]. In previous studies, we identified a subpopulation of healthy rats whose primary deficit consists of less advantageous strategies of choice under uncertain and complex conditions of choice as tested in the Rat Gambling Task (RGT) [9,10]. Healthy poor decision makers are consistently found between labs [11,12], across strains (WH, DA [9] and SD (this paper and [11,12])), and across species (mouse [13], primate and monkey [14]). In rats, poor decision makers present a unique combination of preserved and impaired neurological and behavioral characteristics [9,10,15,16,17]. They express high reward-seeking and risk-seeking behavior, inflexibility, and a tendency to dominant behavior together with normal control of cognitive impulsivity, economical risk taking, and a normal social everyday life [9,16]. In addition, poor decision makers exhibit an imbalance in brain monoaminergic neurotransmitters and a smaller and weaker cortical–subcortical brain network activated during the RGT [18]. With such a vulnerable profile it is, however, not known if poor decision makers are indeed more vulnerable to an acute biological change in the way that it would impair their phenotype more than the phenotype of other individuals.

Central serotonin is an essential neuromodulator for mental health and a promising transdiagnostic marker of mental illness. Selective serotonin reuptake inhibitors can be prescribed following adverse life events in order to boost the central serotonergic system and improve coping processes when facing, for example, grief, unsolicited work termination, or seasonal affective disorder [19,20,21]. In animals, the models of choice to cause an acute biological perturbation are genetic On–Off inducible models [22]. The new TetO-shTPH2 transgenic rat [23,24] is a knockdown model that targets serotonin synthesis to create a mild acute drop in central serotonin. The application of Doxycycline (Dox) in the drinking water of TetO-shTPH2 rats induces the expression of shRNAs against messenger RNA of tryptophan hydroxylase 2 (TPH2), which results in a decrease of up to 25% in brain serotonin levels [23]. In this study, we used TetO-shTPH2 rats to test the impact of acute moderate brain serotonin imbalance on cognition and depending on individual spontaneous decision making type. We focused on complex decision making and risky based decision making, cognitive flexibility, and social recognition memory which are serotonin-dependent transdiagnostic symptoms of psychiatric disorders. We explored how these functions interacted with each other using a novel behavioral network analysis. Following our hypothesis of increased vulnerability in poor decision makers, we expected the moderate drop in serotonin function to impair more specifically their behavior and that it would also reflect on the properties of their behavioral network.

## 2. Results

In this study, we performed a battery of behavioral tests comparing Dox-treated Te-tO-shTPH2 rats (Tph2-kd) rats with a control group (Tph2-wt) consisting of Dox-treated wild-type Sprague Dawley (SD) rats and untreated TetO-shTPH2 (TetO-water) rats (Figure 1).

In both groups, each animal showed either one of the three typical decision-making strategies observed in the RGT (Figure 2A). All animals started the test at 50% of advantageous choices (Figure 2A, Kruskal–Wallis rank sum test, chi-squared = 10.2, df = 5, *p*-value = 0.067, after 10 min: Tph2-wt-gdm 56 + 17 (mean + sd), Tph2-wt-int 52.5 + 17, Tph2-wt-pdm 38.5 + 24, Tph2-kd-gdm 59.6 + 17, Tph2-kd-int 52 + 11, Tph2-kd-pdm 39.2 + 18). For good decision makers (upward triangles) the percentage of advantageous choices increased until reaching a very high preference at the end of the test (above 70%). For poor decision makers (downwards triangles) the percentage of advantageous choices decreased until reaching a very low preference at the end of the test (below 30%). For intermediate individuals (squares) the percentage of advantageous choices stayed at approximately 50% until the end of the test (between 70% and 30%). More disadvantageous choices (RGT score < 30%) were made by Tph2-kd rats than by Tph2-wt rats (Figure 2B left, Wilcoxon rank sum test, W = 2112.5, *p*-value = 0.045). The difference between the Tph2-wt and Tph2-kd groups was found in the proportion of individuals in each decision-making category, with a higher proportion of poor decision makers in Tph2-kd than the Tph2-wt group (Figure 2B right, Fisher’s exact test, *p*-value = 0.044). The distribution density of each group similarly illustrated the decrease in individuals showing good decision making strategy after an acute serotonin drop in Tph2-kd animals (Figure 2B right). The latency to collect the reward was dependent on the decision-making group (Appendix A, p.anova, RGT score, F(1,115) = 38, *p*-value < 0.001) and independent of the treatment group (Appendix A, p.anova, treatment, F(1,115) = 0.14, *p*-value < 1.103). Behavioral flexibility was similarly expressed in Tph2-wt and Tph2-kd animals (Figure 2C). Good decision makers and intermediate decision makers expressed low to high flexibility indexes, whereas poor decision-making animals of both groups expressed low flexibility indexes (Figure 2C).

The decreasing probability of obtaining the large reward induced a discounting effect on the preference for the large reward (Figure 3A left, p.anova, *probability*, F(4,256) = 153, *p*-value < 0.001). This discounting effect was stronger in the low-serotonin poor decision-maker group than in the other groups, as shown by the area under the curve (AUC), which was lower for the low-serotonin poor decision maker group (Figure 3A right, Kruskal–Wallis rank sum test, chi-squared = 13.02, df = 5, *p*-value = 0.023). All groups showed interest in the social partner (social preference, SP), as indicated by the increase in the interaction time during the first social encounter (E1) compared to habituation (Hab, Figure 3B left). Rats also formed a short-term social recognition memory (STM) of the social partner, indicated by the decrease in interaction time between E1 and the third encounter (E3, Figure 3B left, p.anova, *encounter*, F(3,234) = 202, *p*-value = 0). However, low-serotonin poor decision makers showed a lower decrease in interaction time from E1 to E3 (Figure 3B left, p.anova, *encounter* × *treatment* × *RGT*, F(3,234) = 3, *p*-value = 0.018) and the STM ratio for this group was not different from 1, unlike the other groups (Figure 3B right, Wilcoxon signed rank test with continuity correction, V = 9, *p*-value = 0.786), indicating a lack of social recognition. The odor discrimination ability was similar between the Tph2-wt, Tph2-kd, and decision making subgroups (Appendix A).

We applied a network analysis to the data to understand the relationships between the different functions of the behavioral profile of low-serotonin poor decision makers compared to other Tph2-kd and control animals. In both the Tph2-kd and control groups, without poor decision makers, decision making (RGT score) and motivation for the reward (Lat, i.e., latency to collect reward) were strongly connected (Figure 4A,B). However, other strong pairwise connections differed between groups: in the control group, a strong connection between social preference and short-term recognition memory (SP-STM) was found, while in the Tph2-kd group, a strong connection between short-term recognition memory and motivation for reward (STM-Lat) was found (Figure 4A,B). The network of low-serotonin poor decision makers (n = 5) revealed a central position of STM with strong connections between STM and all other functions (Figure 4C). 

After the behavioral tests, we examined how effectively the Dox treatment reduced serotonin metabolism in Tph2-kd rats. The Dox treatment induced on average a 21% decrease in serotonin levels in the Tph2-kd rats compared to the control group (sd = 23, Figure 5A, Wilcoxon rank sum test with continuity correction, W = 2259, *p*-value < 0.001). The 5-HIAA levels were also decreased by 25% on average (sd = 23, Figure 5B, Wilcoxon rank sum test with continuity correction, W = 2825, *p*-value < 0.001). As expected, the tryptophan levels were stable between groups (Figure 5C, Wilcoxon rank sum test with continuity correction, W = 2034, *p*-value = 0.1141). The ratio of 5-HIAA/TRP indicated a consistent decrease in serotonin metabolism in Tph2-kd animals independent of the duration of the treatment (Appendix A, p.anova, treatment, F(1,107) = 75, *p*-value < 0.001, duration, F(1,7) = 0.23, *p*-value = 0.719) and showed variation in the 5-HIAA/TRP decrease between batches. Due to a technical problem, for batch 12, 5-HT could not be measured correctly; nevertheless, the 5-HIAA/TRP ratio showed the effect of the treatment for these animals (Appendix A).

## 3. Discussion

In this study, an acute and mild reduction of central serotonin in Teto-shTph2 rats resulted in an increased number of individuals making disadvantageous choices in conditions of uncertainty of the RGT. The link between serotonin function and poor decision making was previously shown using equivalent tests of the RGT in systemic dietary or pharmacological approaches [25,26]. In this study, looking at spontaneous individual differences in decision-making strategies, we demonstrated that a reduction in central serotonin levels in previously healthy individuals did not affect all animals uniformly but only a subgroup of them. This discrepancy between studies on the impact of serotonin dysfunction—in all individuals (other studies) versus only some individuals in the population (current study)—could be attributed to the attention we gave here to individual differences as well as the advantages of using the refined model of TetO-shTph2 rats. This rat model offers a temporary, moderate, and physiologically relevant variation in serotonin levels, triggering impairments in the most spontaneously vulnerable individuals. This serotonin drop is brain specific and can be modulated during a specific time window due to the inducible and reversible nature of the model [24]. Therefore, this model prevents the confounding impact of developmental compensatory mechanisms of knock-out models [15] and other off-target effects of classical non-genetic approaches. 

As expected, low-serotonin poor decision makers presented a unique combination of cognitive impairments otherwise preserved in wild-type poor decision makers [9] or in the low-serotonin good and intermediate decision makers. Low-serotonin poor decision makers presented deficits in social recognition memory and probability-based decision making, in addition to the typical hypersensitivity to reward and cognitive inflexibility, traits commonly observed in wild-type poor decision makers [9,16]. This combination of deficits in vulnerable individuals was specific to the acute, moderate, and brain-specific decrease in serotonin function.

In the social recognition test, although low-serotonin poor decision makers exhibited the typical social preference for novel subjects, they maintained a longer investigation time for familiar subjects, indicating a lack of habituation possibly due to a deficit in short-term memory and social recognition of familiarity. Serotonin signaling is known to control social recognition memory [27,28,29,30] and to be critical for the adaptation of social behaviors in the home-cage environment [15,31]. In conditions of degraded serotonin function, vulnerable individuals might present increased difficulties in the integration and transmission of social cues to adjust behavior. In the probability discounting task, at the indifference point (P = 20), although both available options were mathematically equivalent in the total amount of food they provided, low-serotonin poor decision makers, but not the other groups, switched preference for the more certain option where a (small) reward is always delivered. This behavior indicated an increased intolerance to the risk of missing a reward. Interestingly, in the RGT, the poor decision makers’ hypersensitivity to immediate reinforcement was found to be one key driver of their choice in the test [16]. While typically healthy poor decision makers are not impaired in the probability discounting task (PDT) and are able to choose following the absolute amount of reward of each option in this task [9,15], here, associated with a drop in serotonin function, the poor decision makers presented an increased focus on short-term, immediate, and certain rewards vs. long-term rewards. This is in line with the role of serotonin in the anticipation of a future reward and the encoding of reward value [32,33,34,35]. 

Interestingly, despite the known effect of serotonin on behavioral flexibility [36,37,38], in this study behavioral flexibility was not worsened in the low-serotonin group compared to controls for each decision making subgroup. Perhaps the decrease in serotonin function should be more pronounced and/or the reversal task more complex for an effect of serotonin on behavioral flexibility to be evident. It would be interesting to test the Tph2-kd rats in other behavioral flexibility tests, such as reversal learning [39,40], and in more complex and ethological conditions [41] to better understand the specific role of central serotonin in behavioral flexibility. 

Here, we showed that a moderate, realistic, and brain-specific decrease in serotonin levels have a particular effect on a subset of vulnerable individuals only. In these animals, low serotonin levels affected their ability to socially habituate and to make decisions in uncertain (probabilistic) conditions by adopting a “short term” strategy focused on immediate and certain gains and in line with an increase in sensitivity to rewards. Considering the rewarding effect of social interactions [42], beyond impaired recognition, sustained interest for a social partner could reflect a generalization of high motivation for reward from food to social interactions. The role of serotonin in modulating social and non-social cognition is well documented [25,43]. However, it is not yet known on which function(s) serotonin could have a primary impact and if impairments then propagate to other functions. Considering the centrality of social recognition memory in the behavioral network of the low-serotonin poor decision makers, an alternative view of our results could be that serotonin alters primarily social cognition and, by diffusion, alters the other connected cognitive functions within the network, especially risky probability-based decision making. This would not be the first time that the importance of studying social cognition as a potential origin point for the modulation of other executive and seemingly non-social functions has been pointed out. In humans, for instance, network approach studies have emphasized the importance of social cognition for executive function and the ability to perform daily essential activities in schizophrenia patients [44,45]. Although we did not assess decision making before inducing the decrease in serotonin, it could be assumed that a pre-existing vulnerability of the poor decision making profile could have made them more sensitive to the decrease in serotonin function and induce the impairments seen in this study. This could explain why other low-serotonin non-poor decision makers were not similarly affected by the mild central serotonin decrease. Also, following our exploratory network analysis, we propose to further investigate the properties of the behavioral network of poor decision makers in normal and “pathological” low-serotonin conditions. Longitudinal experiments in a group-housed complex semi-natural environment, for example, will increase the ethological validity of the behavioral network analysis by informing about the temporal relationships between specific traits of poor decision makers. This would contribute to the understanding of the role of social cognition in the transition to a pathological state, as a target to prevent the establishment of a psychopathology and challenge the role of serotonin as a triggering factor of psychopathology contributing to the current debate on the serotonin hypothesis of psychiatric disorders [46,47,48]. Further studies should apply the combination of the differential, multidimensional, and genetic approaches presented here to model, at the individual level, the transition phenomenon from an adaptive to a pathological state and to reveal the emerging markers of pathology in spontaneously vulnerable individuals. 

In this work, some limitations can be pointed out. First, we only studied male rats. Sex differences in decision making, however, exist in both animals [49,50] and humans with strong serotonergic neurobiological correlates [51]. It is of prime importance to extend our study to female rats in order to evaluate the impact of a serotonin decrease on the cognitive functions at the group level and on the interaction between functions, especially in female “low-serotonin poor decision makers”. It was shown recently that sex differences in decision making may be underpinned by distinct cognitive mechanisms [52], which are critical for the study of vulnerability. Second, levels of 5-HT, 5-HIAA, and tryptophan could vary between batches of animals (Appendix A), although procedures were implemented to reduce variability. Those procedures are reported in the Materials and Methods. Thirdly, administering the drug in drinking water is not the most precise method, but it remained the best option for the chronic behavioral study we present here. To ensure the dose of Dox remained as consistent as possible, we adjusted the concentration of Dox according to the actual body weight and drinking volume of the animals.

## 4. Materials and Methods

### 4.1. Animals

We used 96 male TetO-shTPH2 transgenic rats of Sprague Dawley background (SD) and 24 male SD rats (Figure 1). Control animals (Tph2-wt) included 36 TetO-shTPH2 rats treated with water (TetO-water) and 24 SD rats treated with Dox (SD-Dox). Tph2-knockdown (Tph2-kd) group included the 60 TetO-shTPH2 rats treated with Dox (TetO-Dox). Number of animals for each test and exclusion criteria are reported in Figure 1. Animals were tested by group of twelve individuals called a batch and consisting of six controls and six Tph2-kd. Eight out of ten batches of animals were tested at the same time in pairs. Pairs of batches are indicated in Appendix A. 

Animals were born at the Max Delbrück Center for Molecular Medicine, Berlin, and transferred to the animal facility of Charité—Universitätsmedizin Berlin between nine and ten weeks of age. They were housed in standard rat cages (Eurostandard Type IV, 38 cm × 59 cm) in pairs of animals receiving the same treatment (TetO-water, SD-Dox or TetO-Dox). Cages were maintained in temperature-controlled rooms (22 °C–24 °C and 45%–55% humidity) with inverted 12 h light–dark cycles. Animals had ad libitum access to water and to standard maintenance food (V1534-000, Ssniff, Soest, Germany). During operant training and testing, they were maintained at 95% of their free-feeding weight. After their daily operant testing, rats were fed up to 40 g per animal depending on the amount of reward (sweet pellets) they received in the operant chamber and following an unpredictable schedule (one to several hours after the end of test) to avoid their anticipation of feeding. Rats were weighed every two to three days allowing for adjustment of their portion of standard food and drug treatment.

### 4.2. Treatment

Animals received Dox from the first day of operant training and until the end of the protocol in the drinking water of the cage (Figure 1). The Dox solution was prepared every three days at a dosage of 40 mg/kg of body weight. The concentration of the solution was adapted to the average water consumption and body weight of the animals. For the batches tested in pairs, Dox solutions were prepared from the same supply bottle by the same experimenter and at the same time to apply the same conditions to both batches.

### 4.3. Sacrifice and Brain Collection

Two days after the last test, whatever the length of the protocol, rats were anesthetized via an intraperitoneal injection of Ketamine (100 mg/kg) and Xylazine (10 mg/kg) under isoflurane anesthesia. Pairs of batches were sacrificed by the same experimenters over two days. Animals were transcardially perfused with phosphate-buffered saline. Brain parts were immediately collected, snap-frozen on dry ice, and stored at −80 °C until further use. Orbitofrontal area, prefrontal area, hippocampus, hypothalamus, and raphe were dissected. Brain tissue was weighed after freezing. After the first pilot batches tested in pair (11 and 12), the time of the sacrifice was controlled to prevent circadian effects on the measurements. Sacrifice started one hour after the start of the dark phase except for batch 12, for which it started five hours before the dark phase. 

### 4.4. HPLC Analysis

For the determination of the dosages of monoamines and their metabolites in brain tissue, frozen tissues were homogenized in 300 µL lysis buffer containing 10 µM ascorbic acid and 1.8% perchloric acid using a FastPrep system (VWR, Darmstadt, Germany). Samples were centrifuged for 30 min at 13,000 rpm. Supernatants were transferred in Eppendorf tubes and stored at −80 °C until HPLC measurement. Tissue levels of TRP, 5-HT and its metabolite 5-HIAA were analyzed using high sensitive HPLC with fluorometric detection (Shimadzu, Tokyo, Japan). Sample separation took place at 20 °C on a C18 reversed-phase column (OTU LipoMareC18, AppliChrom Application & Chromatography, Oranienburg, Germany) using a 10 mM potassium phosphate buffer, pH 5.0, containing 5% methanol with a flow rate of 2 mL min^–1^. 

Calculation of substance levels was based on external standard values. Amounts of 5-HT, 5-HIAA, and TRP were measured in hypothalamic samples and normalized to the wet tissue weight for statistical analysis. Individual concentrations of 5-HT, 5-HIAA, and TRP were normalized per batch to the mean of the control individuals and are presented as percentage.

### 4.5. Behavioral Testing

Animals were grouped in batches of 12 animals (6 Tph2-wt and 6 Tph2-kd) and were tested either in the morning or in the afternoon (i.e., 24 animals per day) depending on the light cycle of the housing room (lights on at 20:00 in room 1 or 01:00 in room 2) in order to maximize the use of our four operant cages and minimize potential circadian effect. Rats were all tested 1 h after start of dark phase and within less than 3 h (5 h for the social recognition test).

#### 4.5.1. Operant System

The four operant cages (Imetronic, Marcheprime, France) contained on one side a curved wall equipped with two or four nose-poke holes, depending on the test. On the opposite wall was a food magazine connected to an outside pellet dispenser filled with 45 mg sweet pellets (5TUL Cat#1811155, TestDiet, St. Louis, MO, USA). A separator with a 10 × 10 cm aperture was placed in the middle of the cage. The same light conditions were applied to the four cages.

#### 4.5.2. Rat Gambling Task (RGT)

For the RGT, operant cages were equipped with four nose-poke holes arranged in two pairs (5 cm between holes) on each side of the curved wall (12.5 cm between pairs). The training and testing procedures [9,10,15,16] were adapted to the treatment period required by the tetO system. The training started at day 1 of the Dox treatment (Figure 1). First, rats learned to poke into the nose-poke holes and retrieve the associated reward (1 pellet) into the magazine. Training 1 was completed when 100 pellets had been collected in 30 min (cut-off). Training 2 consisted of poking two consecutive times into the same hole to obtain a reward (1 pellet). Training 2 was completed when 100 pellets had been collected in 30 min (cut-off). Two consecutive nose-pokes into the same hole were considered as a choice for all operant tests. Then, training 3a consisted of a short session during which a choice to any hole was rewarded with 2 pellets within 15 min (cut-off and 30 pellets maximum). After that, a forced training [9] was given to those rats who had a preference above 60% for one of the two pairs of holes in the last session of training 2. During the first part of the forced training, the two nose-poke holes on the non-preferred side were active and lit; the two holes on the preferred side were inactive and not lit. Choosing the active holes induced the delivery of one pellet. After the collection of 15 pellets, the second part of the forced training started with the four holes active and lit. Choosing holes of the formerly preferred side induced the delivery of one pellet with a probability of 20%, whereas choosing the formerly non-preferred side induced the delivery of one pellet with a probability of 80%. The cut-off was 50 pellets or 30 min. The whole training was completed in seven to ten days. Rats were fed ad libitum from training completion until day 18 of the treatment (Figure 1). On day 19, a session of training 2 was performed in order to check the behavior of the rats and any side preference. A forced training was applied on the same day to the rats with a preference for one side superior to 80% (n = 4 animals). This criterion was refined depending on which side the preference was developed for the last three cohorts, the preference threshold was 80% for the side that would be advantageous during the test and 70% for the side that would be disadvantageous (n = 3 animals).

Testing took place on the twentieth day of treatment for 60 min. Two nose-poke holes on one side were rewarded with a large reward (two pellets) and associated with unpredictable long time-outs (222 s and 444 s with the probability of occurrence ½ and ¼, respectively). This was the disadvantageous option, leading to a lower maximum gain of pellets in 60 min. Two nose-poke holes on the other side were rewarded by one pellet and associated with unpredictable short time-outs (6 s and 12 s with the probability of occurrence ½ and ¼, respectively). This was the advantageous option, leading to a maximal gain of pellets within 60 min. The percentage of advantageous choices for the last 20 min of RGT was used to identify the sub-types of decision makers: good decision makers (GDMs) with more than 70% of advantageous choices, poor decision makers (PDMs) with less than 30% of advantageous choices, and intermediate animals in between. Computed over the 60 min of testing, the percentage of advantageous choices per ten-minute interval indicated the progression of the preference over time. An index of the motivation for the reward was measured as the mean latency to visit the magazine after a choice.

#### 4.5.3. Reversed-RGT

The animals were tested in the reversed-RGT 48 h after the RGT. For this test, the two disadvantageous options were spatially switched with the two advantageous options [9,15,16]. A flexibility score was calculated as the preference during reversed-RGT for the location of the non-preferred option during the RGT. Flexible rats had more than 60% of such choices during the last 20 min, undecided rats had between 60% and 40% choices, and inflexible rats had less than 40% choices.

#### 4.5.4. Probability Discounting Task (PDT)

For the PDT, operant cages were equipped with two nose-poke holes on each outer side of the curved wall (25 cm between holes). The protocol was adapted to reduce its overall duration [15] due to the pharmacological treatment. One hole (NP1) was associated with a small and sure reward (1 pellet), the second one (NP5) was associated with a large and uncertain reward (5 pellets) [25]. The delivery of the large reward was dependent on the probability applied that changed from high to low during the experiment: P = 1, 0.33, 0.20, 0.14, and 0.09. The probability was fixed for a day and increased the next day only after reaching the stability criterion. At least three training sessions (P = 1) were performed and a percentage of choice of the large reward ≥70% on two following sessions with ≤15% variation was required to start the test. During testing sessions (P < 1), the stability criterion was ≤10% variation of choice of the large reward during two consecutive sessions. The percentage of preference for the large and uncertain reward was calculated for each probability as the percentage of NP5 choices during the two stable sessions. To calculate the area under the curve (AUC), which represents the sensitivity to probabilistic uncertainty and risk taking, for each individual the preference for the large reward for each probability was normalized to the preference for the large reward during the training (P = 1) and plotted versus the probabilities expressed as odds, with odds = (1/P) − 1 [53]. 

#### 4.5.5. Social Recognition Task (SRt) [15]

The test took place in a square open field (50 × 50 cm). A small clean and empty cage was placed in one corner of the open field shielded by walls to avoid the test rat hiding behind the cage. The unfamiliar conspecifics were older Wistar Han rats, accustomed to the procedure. A video camera on top of the OF recorded the experiment. The subject was placed in the OF containing the empty cage for a habituation of 15 min. Then, the unfamiliar conspecific was placed in the small cage and the subject was allowed to freely explore the open field for five minutes (E1). After that the small cage with the conspecific was removed from the open field, a second clean and empty cage was used to fill the space, and the subject remained alone in the open field for a break of 10 min. The encounter procedure was repeated two more times with the same conspecific (E2, E3). The time spent in close interaction with the unfamiliar rat was measured for each encounter and for the first five minutes of Habituation (Hab) when the subject smelled at the grid of the empty cage. The social preference was calculated as the ratio of the interaction time in E1 and Hab. The short-term social recognition memory was calculated as the ratio of the interaction time in E1 and E3.

#### 4.5.6. Odor Discrimination Test [15]

The test took place in a square OF (50 × 50 cm). Two plastic petri dishes filled with either used or fresh bedding were placed in two opposite corners of the OF. Except for the first group, the dishes were taped onto the OF flour to avoid animals pushing them around. A video camera above the OF recorded the experiment. The test rat explored the OF for 5 min. The time spent in close interaction with each dish was measured by trained observers using JWatcher (version 1.0) [54] and the preference for the used bedding (social odor) was calculated.

### 4.6. Statistical Analysis

The free software R (version R-3.6.1) and R studio (version 1.1.456) were used for the statistical analyses [55]. We used the Wilcoxon rank sum test to compare the Tph2-wt and Tph2-kd groups, the Kruskal–Wallis rank sum test to compare the decision maker groups against each other (GDMs, INTs and PDMs of Tph2-wt and Tph2-kd groups), Fisher’s exact test to compare the number of GDMs and PDMs in each groups, and the Wilcoxon sign test (Package RVAideMemoire) [56] to compare the performance of the animals to a theoretical value in PDT, SRt, and odor discrimination test. We used ANOVA with permutations (package lmPerm) [57], which is suited for small groups and non-parametric data to compare the multiple groups and decision maker groups (dm-group) over several time points (probability, encounter), with animal as error factor. 

### 4.7. Network Visualization of Behavioral Data

We used a network analysis method (Package qgraph) [58] to visually represent the strength of connections between pairs of behaviors, as behavioral networks of the groups and subgroups. According to the network approach of psychopathology, the visual representation of the connectivity between symptoms could inform about the potential dynamic existing between them in pathological context. In this study, we used this visualization to explore how the five tested cognitive functions connected to each other depending on their treatment and decision making profile. We were also interested to see if one function would appear to be more central (in terms of number and weight of ties to other nodes) than the others.

The five parameters constituting the nodes of the networks were the RGT score, mean latency to visit the magazine after a choice during RGT, AUC of the PDT, social preference, and short-term social recognition memory. They represented complex decision-making ability, motivation for reward, risky decision-making ability, social preference, and social recognition memory, respectively. The reversed-RGT and odor discrimination tests were not performed by all animals and were not included in the networks. The network analysis was performed using the qgraph() function (type of graph “association”) where correlations are used as edge weights between two nodes. Although partial correlations are usually preferred to correlations as they account for the relationships of the network for each pairwise link, it was not possible in our analysis to apply Spearman’s partial correlation to the Tph2-kd poor decision maker group as the number of individuals equaled the number of functions. We used the Spearman’s correlations (package Hmisc) for all groups. To simplify the description and visualization of the networks, only correlations above 0.28 were represented, and the networks were calibrated from 0 to 0.7. We calculated the strength centrality for each node, which is the centrality of a node taking into account the number and the weight of edges connecting to the node [59]. With only one control poor decision maker individual, no network could be computed for this subgroup.

## 5. Conclusions

The destabilization of the serotonergic system, using the TetO-shTph2 model, led to specific cognitive and social impairments in only a subset of vulnerable individuals from the general population. Besides their poor abilities to choose long-term advantageous feeding options, the low-serotonin poor decision makers expressed poor social recognition memory and a strong risk aversion in a probability-based decision-making task, whereas the individuals with a more adapted strategy were not affected by the temporary serotonin decrease in any other tests. The behavioral network of low-serotonin poor decision makers was deeply affected by the decrease in central serotonin. Taken together, these findings may suggest that social recognition memory is a key factor dependent on serotonin function and is at the core of a larger cognitive network. With this study, we show the high potential of the Teto-shTph2 rat to model in further specific studies the transition processes of psychopathological deterioration associated with a central serotonin drop. 

## Figures and Tables

**Figure 1 ijms-25-05003-f001:**
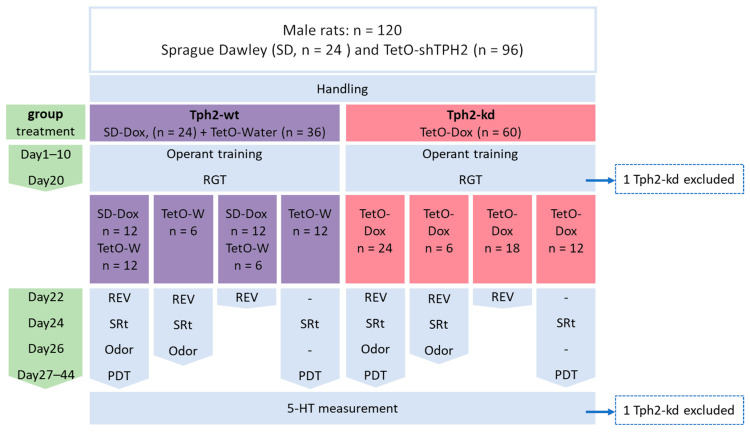
Flow chart for reporting attrition and experimental design. Dox treatment started at Day1 with operant training. Five behavioral tests were run: Rat Gambling Task (RGT) with n = 60 Tph2-wt and n = 60 Tph2-kd; reversed-RGT (REV) with n = 48 Tph2-wt and n = 48 Tph2-kd; social recognition task (SRt) with n = 42 Tph2-wt and n = 42 Tph2-kd; odor discrimination test (Odor), a control measure of the olfactory ability with n = 24 Tph2-wt and n = 24 Tph2-kd; and probability discounting task (PDT ) with n = 36 Tph2-wt and n = 36 Tph2-kd. Two Tph2-kd rats were excluded from the study: one animal excluded from RGT, reversed-RGT, and subsequent decision-making subgroups analysis because it did not perform correctly in the RGT (its motivation was low and it paused during the test; its pattern of choice did not fit any known pattern); and one animal was excluded from the whole analysis because the serotonin (5-HT) level was higher than mean + 2 standard deviations.

**Figure 2 ijms-25-05003-f002:**
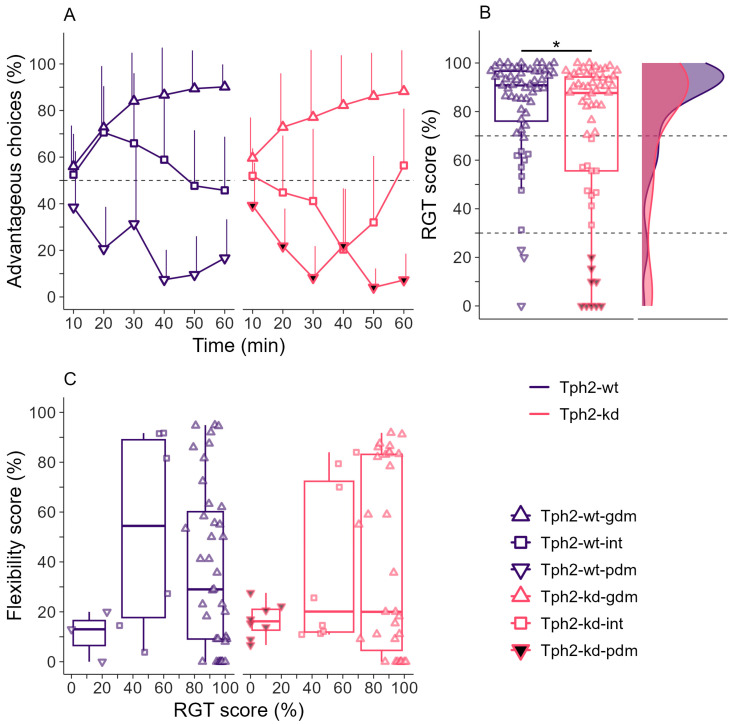
(**A**) Advantageous choices in the Rat Gambling Task (RGT) over time (10 min bins). Lines indicate mean + sd, dashed line shows 50% chance level. (**B**) **Left:** Individual (mean) scores during the last 20 min of the RGT (RGT score). Wilcoxon rank sum test, * *p*-value < 0.05. The dashed lines at 70% and 30% of advantageous choices visually separate good decision makers (gdm, upward triangle, above 70% of advantageous choices in the last 20 min), intermediates (int, square, between 30S% and 70% of advantageous choices in the last 20 min), and poor decision makers (pdm, downward triangle, below 30% of advantageous choices in the last 20 min). Individual data over boxplots. **Right:** Distribution density of final scores for Tph2-wt and Tph2-kd groups. (**C**) Flexibility scores in the reversed-RGT corresponding to the preference for the new location of the preferred option in the RGT for gdm (upward triangle), int (square), and pdm (downward triangle). The flexibility score is the preference for the location of the non-preferred option during the RGT. Individual data over boxplots. Boxplots classically represent the median, 25th and 75th percentiles, and 1.5 IQR. Panels (**A**,**B**): Tph2-wt, n = 60 (gdm, n = 49, int, n = 8, pdm, n = 3), Tph2-kd, n = 58 (gdm, n = 39, int, n = 10, pdm, n = 9). Panel (**C**): Tph2-wt, n = 48 (gdm, n = 39, int, n = 6, pdm, n = 3), Tph2-kd, n = 47 (gdm, n = 31, int, n = 8, pdm, n = 8).

**Figure 3 ijms-25-05003-f003:**
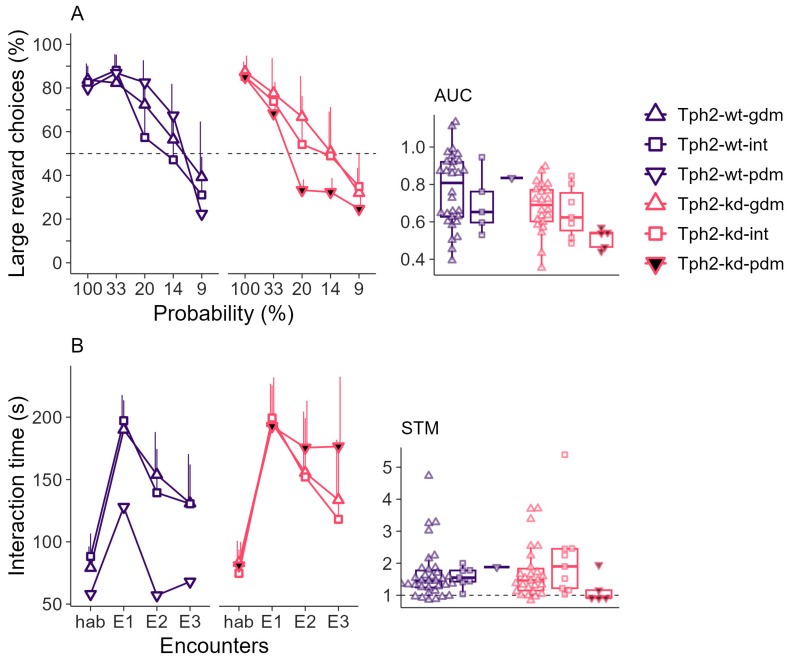
(**A**) **Left:** Choice of the large reward option as a function of the probability of reward delivery in the probability discounting task (PDT). Lines indicate mean + sd, dashed line shows 50% chance level. **Right:** Area under the curve (AUC) per individual. (**B**) **Left:** Duration of interaction in the social recognition task (SRt). Lines indicate mean + sd. Habituation with empty cage (Hab), successive encounters with same conspecific placed in the small cage (E1–3). **Right:** Short-term memory ratio (STM) between E1 and E3. Boxplots classically represent the median, 25th and 75th percentiles, and 1.5 IQR. Panel (**A**): Tph2-wt, n = 36 (gdm, n = 30, int, n = 5, PDMs, n = 1), Tph2-kd, n = 34 (gdm, n = 22, int, n = 7, pdm, n = 5). Panel (**B**): Tph2-wt, n = 42 (gdm, n = 34, int, n = 7, pdm, n = 1), Tph2-kd, n = 40 (gdm, n = 26, int, n = 9, pdm, n = 5).

**Figure 4 ijms-25-05003-f004:**
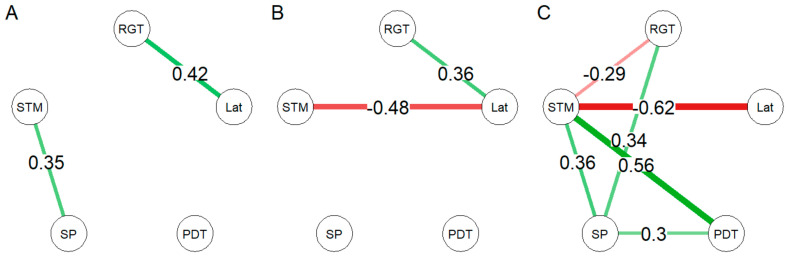
Network analysis of the behavioral profiles with Spearman’s correlations. (**A**) Tph2-wt and (**B**) Tph2-kd groups without poor decision maker animals and (**C**) low-serotonin poor decision makers (from the Tph2-kd group). Edges between cognitive functions are Spearman’s correlations, green edges for positive correlations and red edges for negative correlations. Only strong correlations (r > 0.3) are indicated and thickness of the edge the strength of the correlation. RGT: Decision-making ability; Lat: motivation to collect reward; PDT: impulsive choices; SP: recognition of social novelty; STM: recognition of social familiarity. Panel (**A**): Tph2-wt without poor decision makers, n = 35. Panel (**B**): Tph2-kd without poor decision makers, n = 29. Panel (**C**): low-serotonin poor decision makers, n = 5.

**Figure 5 ijms-25-05003-f005:**
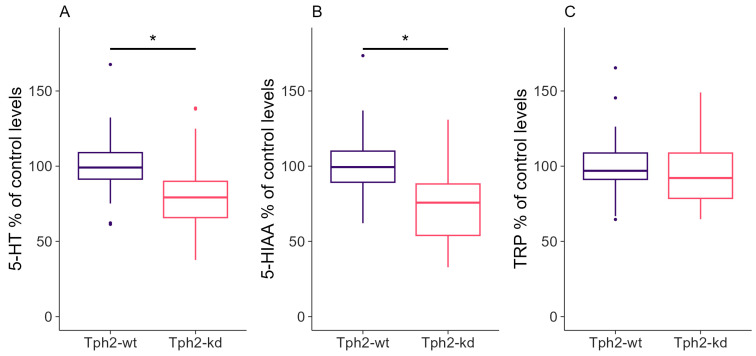
Levels of (**A**) serotonin (5-HT), (**B**) 5-HIAA, and (**C**) Tryptophan (TRP) in the hypothalamus. Values were normalized to control (Tph2-wt) individuals within each batch; one batch was excluded from (**A**) due to a technical problem in serotonin detection. Boxplots classically represent the median, 25th and 75th percentiles, and 1.5 interquartile range (IQR). * *p*-value < 0.05. Panel (**A**): Tph2-wt, n = 54; Tph2-kd, n = 54. Panels (**B**,**C**): Tph2-wt, n = 60; Tph2-kd, n = 59.

## Data Availability

The original data presented in the study and scripts are openly available at https://github.com/alonsolucille/TetO-shTPH2_rats.

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
