# Peer review of "Poor Decision Making and Sociability Impairment Following Central Serotonin Reduction in Inducible TPH2-Knockdown Rats"

_ijms, 2024, doi:10.3390/ijms25095003_

Round 1

Reviewer 1 Report

Comments and Suggestions for Authors

Dear authors, I have read the manuscript entitled "Poor decision making and sociability impairment following central serotonin reduction in inducible TPH2-knockdown rats". The results of this original, preclinical study conducted on laboratory animals (inducible TPH2-knockdown rats) show that destabilization of the serotoninergic system leads to specific social cognitive disorders.

The paper complies with the requirements of the journal, the obtained results were highlighted with the help of figures, giving the reader the possibility to make the reading easier. Bibliographic references used for documentation are in accordance with the chosen topic. Although it wasn't mandatory, I appreciate that you also gave us some takeaways.

However, I have a few questions and suggestions:

1. The percentage of similarity is quite high (87%). I know that for the established terms, you don't really have a choice, but try to rephrase so that the paper can meet the requirements of the journal.

2. I noticed that you used doxycycline administration in the drinking water. You have evaluated its result by dosages. However, such administration raises doubts about ensuring the appropriate dose. Can the authors provide some clarification on this?

3. Did the two anesthetics (ketamine and xylazine) administered as general anesthetics for slaughter affect the final results?

4. For figures 1, 2 and 4 I recommend that the writing be legible.

5. For the Conclusions section, I ask the authors to formulate clear ideas. At the end, a series of references also appear. If the conclusions are from this study, why must references appear? Do these conclusions belong to other authors?

6. What are the limitations of this study?

7. English can be improved.

Comments on the Quality of English Language

English can be improved.

Author Response

Reviewer 1

Dear authors, I have read the manuscript entitled "Poor decision making and sociability impairment following central serotonin reduction in inducible TPH2-knockdown rats". The results of this original, preclinical study conducted on laboratory animals (inducible TPH2-knockdown rats) show that destabilization of the serotoninergic system leads to specific social cognitive disorders.

The paper complies with the requirements of the journal, the obtained results were highlighted with the help of figures, giving the reader the possibility to make the reading easier. Bibliographic references used for documentation are in accordance with the chosen topic. Although it wasn't mandatory, I appreciate that you also gave us some takeaways.

 However, I have a few questions and suggestions:

  1. The percentage of similarity is quite high (87%). I know that for the established terms, you don't really have a choice, but try to rephrase so that the paper can meet the requirements of the journal.

We tried our best to rephrase the text. The changes are marked in the revised version of the manuscript. Our Material and Methods section may match our previous articles as it is not always possible to reformulate the detailed procedures.

  1. I noticed that you used doxycycline administration in the drinking water. You have evaluated its result by dosages. However, such administration raises doubts about ensuring the appropriate dose. Can the authors provide some clarification on this?

We agree with the reviewer that administering the drug in the drinking water is not the most precise method, but it remains the best option for the chronic behavioral study we present here. To ensure the dose remained as consistent as possible, we adjusted the concentration of DOX according to the actual body weight and drinking volume of the animals, as detailed in the materials and methods section. We added this idea in the limitations of the study (see 6.).

  1. Did the two anesthetics (ketamine and xylazine) administered as general anesthetics for slaughter affect the final results?

Ketamine and xylazine are classical anesthetics used to sacrifice rodents for scientific purposes. Xylazine, being an alpha-2 adrenergic receptor agonist, is very unlikely to have direct effects on the serotonergic system. Ketamine primarily acts as an NMDA receptor antagonist and was also discussed to interact with monoamine transporters. However, subsequent studies have demonstrated that ketamine and its metabolites do not exhibit inhibitory effects on neurotransmitter transporter activity, including SERT, at concentrations up to 10 μM (Can et al., 2016). Considering that the same procedure was applied to all animals, the impact of the anesthesia on tissue serotonin levels is very unlikely.

Can A, Zanos P, Moaddel R, Kang HJ, Dossou KS, Wainer IW, Cheer JF, Frost DO, Huang XP, Gould TD. Effects of Ketamine and Ketamine Metabolites on Evoked Striatal Dopamine Release, Dopamine Receptors, and Monoamine Transporters. J Pharmacol Exp Ther. 2016; 359(1):159-70. doi: 10.1124/jpet.116.235838.

  1. For figures 1, 2 and 4 I recommend that the writing be legible.

We have improved the readability of the figures increasing font size and figure size of all the figures.

  1. For the Conclusions section, I ask the authors to formulate clear ideas. At the end, a series of references also appear. If the conclusions are from this study, why must references appear? Do these conclusions belong to other authors?

We have moved the perspective sentence to the discussion (line 296-303) and have modified the conclusion to improve clarity.

  1. What are the limitations of this study?

We have included in the discussion (line 308) a section reporting the limitations of the study.

  1. English can be improved.

We have carefully reviewed the manuscript and improved the English language. We thank the reviewer for their useful comments.

Reviewer 2 Report

Comments and Suggestions for Authors

Alonso et al. present a novel TPH2-knockdown rat model to assess the impact of reduced serotonin metabolism in decision making. By administering the antibiotic doxycycline orally, this model has a reduction in serotonin production and experimentation with classic behavioral tests could be conducted.

First, the authors demonstrated a reduction in serotonin production and metabolite generation to demonstrate the magnitude of effect in their model, as displayed in figure 1.

What is confusing is the presentation of the biochemical results obtained from sacrificed animals after they underwent behavioral tests over two days. What is also confusing is that the methods Section seems to have 24 normal SD rats combined with 36 TPH2-knockdown rats administered water compose the “control group”, with another 60 TPH2-knockdown rats administered doxycycline. Figure 1 only mentions “Tph2-wt”, water treated animals, with n=54-60. The experimental flow figure presented in methods should be presented in Results, and the conflicts in animal number and type resolved.

The behavioral test results seem unambiguous, and the statistics  

While perhaps time consuming, a figure that provides the molecular processes studied by the investigation with the specific molecular target of interest clearly identified would nicely frame the following data and conclusions.

Please enlarge figure 1 to enhance readability.

Please make at least 2 separate figures out of figure 2 and enlarge the images. The lack of x-axis for parts of the figures, while somewhat understandable, make the data harder to understand. I recommend x-axis for all graphed data.

Enlarge figure 3 so the letters in the small circles can be more easily read.

Figure 4 should also be enlarged. I do not know what the small groups of letters and numbers towards the bottom of the figure mean (e.g., B11 to B20).If the authors printed out their paper on the usual 8.5-11 inch sheet, they would see how hard these figures can be to interpret. It ruins your interesting message.

In summary, while interesting, the authors need to indicate what sort of animals were used to compose each group. Further, the animals were killed at the end, not the beginning of the work, so the behavioral data should be presented first prior to the brain chemistry results – which were confirmatory. Lastly, the figures need to be enhanced to better transmit the message of the authors.

Author Response

Reviewer 2

Alonso et al. present a novel TPH2-knockdown rat model to assess the impact of reduced serotonin metabolism in decision making. By administering the antibiotic doxycycline orally, this model has a reduction in serotonin production and experimentation with classic behavioral tests could be conducted.

First, the authors demonstrated a reduction in serotonin production and metabolite generation to demonstrate the magnitude of effect in their model, as displayed in figure 1.

What is confusing is the presentation of the biochemical results obtained from sacrificed animals after they underwent behavioral tests over two days. What is also confusing is that the methods Section seems to have 24 normal SD rats combined with 36 TPH2-knockdown rats administered water compose the “control group”, with another 60 TPH2-knockdown rats administered doxycycline. Figure 1 only mentions “Tph2-wt”, water treated animals, with n=54-60. The experimental flow figure presented in methods should be presented in Results, and the conflicts in animal number and type resolved.

As suggested by the reviewer to maintain the logic of the manuscript, we moved the postmortem biochemical data after the behavioral results. Furthermore, we modified the results section to clarify the group composition in addition to their description in the Materials and Methods. We have also modified the flow chart accordingly and moved it to the beginning of the results section.

The behavioral test results seem unambiguous, and the statistics 

Unfinished comment it seems. 

While perhaps time consuming, a figure that provides the molecular processes studied by the investigation with the specific molecular target of interest clearly identified would nicely frame the following data and conclusions.

The molecular processes underlying serotonin reduction in shTPH2-KD rats were described and also presented as a graphical scheme in our manuscript describing this model for the first time (Matthes et al., 2019). The focus of the current paper is the behavioral analysis as a function of serotonin reduction, and not the molecular mechanisms of the model. We are happy to make a graphical abstract linking observed phenotypes to serotonin drop if the journal allows.

Matthes S, Mosienko V, Popova E, Rivalan M, Bader M, Alenina N. Targeted Manipulation of Brain Serotonin: RNAi-Mediated Knockdown of Tryptophan Hydroxylase 2 in Rats. ACS Chem Neurosci. 2019 Jul 17;10(7):3207-3217. doi: 10.1021/acschemneuro.8b00635. Epub 2019 Apr 22. PMID: 30977636.

Please enlarge figure 1 to enhance readability.

Please make at least 2 separate figures out of figure 2 and enlarge the images. The lack of x-axis for parts of the figures, while somewhat understandable, make the data harder to understand. I recommend x-axis for all graphed data.

Enlarge figure 3 so the letters in the small circles can be more easily read.

Figure 4 should also be enlarged. I do not know what the small groups of letters and numbers towards the bottom of the figure mean (e.g., B11 to B20).If the authors printed out their paper on the usual 8.5-11 inch sheet, they would see how hard these figures can be to interpret. It ruins your interesting message.

We have enlarged the Figure 1, 2, 3 and 4. We have increased the readability of all the figures. We have added x-axis on figure 2 and cut the figure in two parts (new figure 2 and figure 3). We have simplified the flowchart figure and improved its legend.

In summary, while interesting, the authors need to indicate what sort of animals were used to compose each group. Further, the animals were killed at the end, not the beginning of the work, so the behavioral data should be presented first prior to the brain chemistry results – which were confirmatory. Lastly, the figures need to be enhanced to better transmit the message of the authors.

We have addressed all these points in the points above. We thank the reviewer for their useful comments.

Round 2

Reviewer 2 Report

Comments and Suggestions for Authors

No further comments.